# Rapid morphological change in black rats (*Rattus rattus*) after an island introduction

Oliver R.W. Pergams[1,2], David Byrn[1], Kashawneda L.Y. Lee[1] and Racheal Jackson[1]

[1] Department of Biology, Olive-Harvey College, One of the City Colleges of Chicago, Chicago, IL, USA
[2] Department of Biological Sciences, University of Illinois at Chicago, Chicago, IL, USA

## ABSTRACT

Rapid morphological change has been shown in rodent populations on islands, including endemic deer mice (*Peromyscus maniculatus* subspp.) on the California Channel Islands. Surprisingly, most of these changes were towards a smaller size. Black rats were introduced to Anacapa Island in the mid-1800s (probably in 1853) and eradicated in 2001–2002. To assess possible changes in these rats since their introduction, eleven cranial and four standard external measurements were taken from 59 *Rattus rattus* specimens collected from 1940–2000. All rat cranial traits changed 3.06–10.43% (724–2567 $d$, 0.06–0.42 $h$), and all became larger. When considered in haldanes, these changes are among the fastest on record in any organism, and far exceed changes found in other island rodents. These changes were confirmed by MANOVA (*Wilk's* $\lambda$ < 0.0005, $F_{d.f.15}$ = 2974.386, $P$ < 0.0005), and all 11 cranial traits significantly fit linear regressions. We speculate that concurrent changes in mice may have been due in part to competition with and/or predation by rats. Future research might evaluate whether the vector of mouse evolution on Anacapa is again changing after rat eradication.

Corresponding author
Oliver R.W. Pergams,
opergams@ccc.edu

Rapid evolution involves changes that occur over <100 years (*Dobzhansky, 1937*; though discussion in terms of generations is more useful). Usually the evolution documented is of the introduced organism itself to its new environment, but sometimes rapid evolutionary response of endemic organisms in reaction to an introduction is also shown (*Hendry & Kinnison, 1999*; *Pergams & Kareiva, 2009*). Introduction (of invasive species and populations) is the most commonly discussed factor in rodent rapid evolution (*Berry, 1964*; *Patton, Yang & Myers, 1975*; *Pergams & Ashley, 2001*; *Pergams, Barnes & Nyberg, 2003*; *Pergams & Lacy, 2007*; *Pergams & Kareiva, 2009*).

Some of the first recognized examples of microevolution came from studies of island rodents (e.g. *Clarke, 1904*; *Huxley, 1942*). An increase in body size is often documented and usually attributed to release of mainland selective pressures (*Case, 1978*; *Foster, 1964*; *Lawlor, 1982*). In a study of tri-colored squirrels in Malaysia, Indonesia, and Thailand,

*Heaney (1978)* shows an inverse correlation of body size with island area up to about 100 km$^2$, above which body size increases. *Lomolino (1985)* shows that trends in body size associated with islands were tied to the animals' absolute size: larger mammals tend to become smaller on islands and smaller mammals tend to become larger.

Comparing cranial and skeletal traits as well as body size, *Pergams & Ashley (2001)* perform a meta-analysis of rapid morphological change in island rodents: in *Mus musculus* after introduction to islands of the North Atlantic (*Berry, 1964*; *Berry, Jakobson & Peters, 1978*); in *Rattus rattus* after introduction to the Galapagos Islands (*Patton, Yang & Myers, 1975*); and in *Peromyscus maniculatus* on the California Channel Islands (*Pergams & Ashley, 1999*; *Pergams & Ashley, 2000*). The authors confirm that microevolution of both gross body size and cranial and skeletal traits are greater on smaller and more remote islands. *Millien & Damuth (2004)*, *Millien (2006)* and *Millien (2011)* suggest that island populations exhibit larger body sizes not because they are evolving toward gigantism, but because their evolution toward smaller size (due to climate warming; *Mayr, 1963*; *Smith et al., 1995*) has merely been slowed.

The eight California Channel Islands, including Anacapa Island, are each home to an endemic subspecies of deer mouse, *Peromyscus maniculatus*. The land vertebrate fauna of the Channel Islands is depauperate, and deer mice are the only land mammal species endemic to all eight islands. Deer mouse populations reach high densities on several of the islands, probably because there are few predators, and seem to exhibit cyclicity in population size, as do voles and lemmings (*Drost & Fellers, 1991*; *Pergams & Ashley, 2000*).

Endemic mice on Anacapa Island (*Peromyscus maniculatus anacapae*) were found to have mostly become smaller (including in body size) between 1940 and 1978, except their noses became broader and their ears became larger (*Pergams & Ashley, 1999*; *Pergams & Ashley, 2000*). These trends are not matched in the other two California Channel Islands evaluated: Santa Barbara and Santa Cruz. Also, there is no record of black rats on either island. The authors suggest responses on Anacapa may be due to the presence of rats.

Black rats (*Rattus* rattus) were likely introduced to the California Channel Islands during the mid- to late 1800s (*Collins, Storrer & Rindlaub, 1979*), probably in 1853 with the shipwreck of the *SS Winfield Scott* on Middle Anacapa islet. The ship was a sidewheel steamer that transported passengers and cargo between San Francisco, California and Panama in the early 1850s, during the California Gold Rush (*Gleason, 1958*). Although mice on Anacapa have been evaluated for rapid change as above, rats on Anacapa have not. Our prediction is that rats will have become larger over time, due to reduced predators and release of selective pressures.

## MATERIALS AND METHODS

A total of 61 black rats from Anacapa Island from the Santa Barbara Natural History Museum, the Natural History Museum of Los Angeles County, and the NGO Island Conservation were examined. We received permission from all museums to access the specimens. Two rats were determined to be juvenile by skull suture (*DeBlase & Martin, 1974*) and excluded from further analysis. The remaining adult specimens and collection

years are as follows: 1940 ($n = 12$), 1975 (5), 1978 (21), 1979 (9), 1986 (2), and 2000 (10). It was not possible to get more modern rat specimens: rats were eradicated on Anacapa in 2001–2002 (*Howald et al., 2009*). Indeed, the ten year 2000 specimens utilized in this study were collected just prior to the eradication. There were 22 males, 16 females, and 21 unknown.

Eleven cranial measurements were taken following *Collins & George (1990)* unless otherwise indicated. Measurements included: alimentary toothrow (AL), breadth of braincase (BB), breadth of rostrum (BR), depth of braincase (DBC), greatest length of skull (GL), interorbital breadth (IB), length of braincase (LBC), length of incisive foramen (LIF), length of palate plus incisor (LPN, measured as the greatest distance from the end of the nasals to the mesopterygoid fossa), length from supraorbitals to nasals (ONL, measured as the least distance from the supraorbital notch to the tip of the nasals), and zygomatic breadth (ZB).

Cranial measurements of a total of 59 specimens were taken by DB, RJ, and KL with digital calipers to the nearest 0.5 mm. Each worker measured each trait three times and utilized the mean. A threshold variance > 0.1 of the three means required re-measuring of that trait. The four standard external measurements in museum specimens were originally made by numerous different museum preparers and recorded from museum tags: total length (TOT), tail length (TAIL), hind foot length (HF), and ear length (EAR). Because of either lack of external measurement by museum preparers or damage to the skulls, some measurements were not available for some specimens. In particular, we did not have external measurements for 23 specimens. In 13 cases this was due to skulls resident in museum collections not having external measurements attached. The 10 specimens from Island Conservation were so highly decayed that any external measurements would have been guesswork. In all cases all measurements available to us were used.

We used visual examination of normal probability plots (*Afifi, Clark & May, 2004*) and the Shapiro–Wilk $W$ statistic to test for normality of distribution. The Kolmogorov–Smirnov $D$ statistic test was not used because these tests have poor power properties and tend to reject the null hypothesis with large sample size and accept it with small sample size (*Afifi & Clark, 1997*).

To evaluate sexual dimorphism, two-sample $t$-tests were performed after dropping 21 specimens of unknown sex (leaving 38 specimens of known sex). Results of $t$-tests were considered significant at the 95% confidence level. The specimens of unknown sex were included in subsequent analyses.

We used two methods to evaluate changes over time: (A) categorical analysis and (B) linear regression Because the longest gap between collection years was 35 years, and the next longest gap was only 14 years, we chose to perform our categorical analysis (MANOVA) on two time periods = 1940 & 1975–2000. Also, these categories fit well with when Anacapa deer mice were collected (1940 & 1978), allowing for more direct comparison. However, damage to specimens and lack of measurements did not permit inclusion of the full multivariate data on many of the specimens. Accordingly, we also performed independent-samples $t$-tests, testing the significance of the difference between

the sample means of the pre- and post-1950 time periods of each measurement of all specimens. We used the Levene statistic (*Brown & Forsythe, 1974*) to test the assumption of equal variance, and applied the appropriate *t*-test. Although there have been recent and substantial objections to the use of sequential Bonferroni corrections (especially by ecologists; *Moran, 2003*), to be conservative we then applied a Holm–Bonferroni sequential correction to account for the multiple tests being conducted (*Holm, 1979*). To compare means of non-normally distributed traits, we used Kruskal–Wallis *H* tests.

Because we wished to evaluate rates of change as well as total amounts of change, we calculated the rate of annual change in each significant trait. This was done by dividing the difference in means by the difference in the means of collection years, both between periods. Darwins were calculated with the equation:

$$d = |(\ln x_2 - \ln x_1)/(t_2 - t_1)|,$$

where $\ln x_1$ and $\ln x_2$ are sample means of ln measurements at times $t_1$ and $t_2$, respectively (measured in millions of years).

However, evolution calculated in standard deviations per generation (haldanes) has several advantages. Rates in haldanes are independent of dimension in a way that rates in darwins are not, since traits are passed onto progeny only between generations, and generation times vary between organisms by many orders of magnitude. This makes haldanes more readily comparable in terms of quantitative evolutionary genetics (*Gingerich, 1993*). Haldanes were calculated with the equation:

$$h = |[(\ln x_2/s\ln x) - (\ln x_1/s\ln x)]/(t_2 - t_1)|,$$

where $\ln x_1$ and $\ln x_2$ are sample means of ln measurements at times $t_1$ and $t_2$ respectively (measured in generations), and $s\ln x$ is the pooled standard deviation of $\ln x_1$ and $\ln x_2$ (*Haldane, 1949*). Published demographic data was used to estimate generations per year (*Erickson & Halvorsen, 1990*).

However, the two time periods compared a very small early (1940) sample (only 12 of the 59 specimens). To lessen this effect, we performed linear regressions comparing each measurement of each trait, with the year in which the specimen was collected.

## RESULTS

Data were first examined for normality of distribution through inspection of normal probability plots (*Afifi, Clark & May, 2004*) and Lilliefors test (*SPSS, 1998*). We found all traits except AL to be normally distributed. There was no significant sexual dimorphism for any of the 15 measurements.

Results of Levene's tests, independent samples *t*-tests, and Kruskal–Wallis tests are given in Table 1. In rats, all cranial measurements (but no external measurements) were found to have changed over time. We feel our not finding significant change in external measurements was very likely due to insufficient sample size; missing from 23/59 (39%) of the specimens. MANOVAs corroborated individual results from the *t*-tests. Rats collected

**Table 1 Test of significant differences between time classes 1940 and 1975–2000. Levene's test was performed to determine equality of variances.** Depending on results, the appropriate t-test was performed to test significance of differences of means between time classes. NND means traits were not normally distributed and non-parametric Kruskal–Wallis H test were performed. ND means traits were normally distributed. Traits with significant changes are shaded.

| TRAIT | Levene's statistic | | Independent samples t-test | | | Kruskal–Wallis $H$ | |
|---|---|---|---|---|---|---|---|
| | F | Sig. | t | df | Sig. | H | Sig. |
| TOT | 0.312 | 0.580 | −1.207 | 36 | 0.235 | ND | ND |
| TAIL | 0.994 | 0.325 | −0.791 | 36 | 0.434 | ND | ND |
| HF | 0.008 | 0.931 | −1.053 | 36 | 0.299 | ND | ND |
| EAR | 8.329 | 0.007 | 0.603 | 34.786 | 0.551 | ND | ND |
| ONL | 0.108 | 0.743 | −2.803 | 49 | 0.007 | ND | ND |
| BR | 0.589 | 0.446 | −2.582 | 54 | 0.013 | ND | ND |
| ZB | 0.543 | 0.465 | −3.013 | 41 | 0.004 | ND | ND |
| IB | 0.428 | 0.516 | −2.634 | 56 | 0.011 | ND | ND |
| BB | 0.496 | 0.485 | −3.2 | 44 | 0.003 | ND | ND |
| LPN | 0.209 | 0.650 | −3.28 | 48 | 0.002 | ND | ND |
| LIF | 0.828 | 0.367 | −2.417 | 52 | 0.019 | ND | ND |
| AL | NND | NND | NND | NND | NND | 6.666 | 0.010 |
| DBC | 3.643 | 0.063 | −3.042 | 40 | 0.004 | ND | ND |
| GL | 0.015 | 0.904 | −2.518 | 43 | 0.016 | ND | ND |
| LBC | 0.049 | 0.826 | −3.041 | 43 | 0.004 | ND | ND |

in 1940 were extremely significantly different from rats collected in 1975–2000 (Wilk's $\lambda < 0.0005$, $F_{d.f.15} = 2974.386$, $P < 0.0005$).

Table 2 gives amounts and rates of evolutionary change. All rat cranial traits changed 3.06–10.43% (724–2567 $d$, 0.06–0.43 $h$), and all became larger. All external traits also grew larger, but probably because of much smaller sample size, not significantly so.

Table 3 and Fig. 1 show the results of our linear regressions. All cranial traits were found to be significant, while none of the external traits were found to be significant.

## DISCUSSION

All rat cranial traits changed, some dramatically, and all became larger. When considered in the more accurate haldanes, some of these changes are among the fastest on record (*Hendry, Farrugia & Kinnison, 2007*), the maximum of 0.43 $h$ (trait IB) exceeded only by systems such as Trinidadian guppy spot number, spot area, spot length, offspring size, & size: 0.43–0.74 $h$ (*Endler, 1980*; *Reznick et al., 1997*); Galapagos finches weight, bill depth, bill width, bill length, beak size: 0.43–0.71 $h$: (*Grant & Grant, 1995*; *Grant & Grant, 2002*); and freshwater copepod egg type switch date: 0.49 $h$ (*Hairston Jr & Walton, 1986*). These rates of change in the rats are far above the highs found in other island rodents, including invasive black rats on the Galapagos Islands and endemic deer mice on the California Channel Islands [0.03 $h$ max (*Pergams & Ashley, 2001*)].

It is interesting that our linear regressions showed all cranial traits to be significant, and that all cranial traits increased monotonically across the three time periods. Four

**Table 2 Calculation of amount and rate of evolutionary change.** Three generations per year was used to calculate haldanes (*Erickson & Halvorsen, 1990*). Shaded areas indicate significant changes, *NS*, not significant.

|  | 1940 Mean | SE | 1975–2000 Mean (1982.894) | SE | Response | % Change | darwins | *haldanes |
|---|---|---|---|---|---|---|---|---|
| ZB | 19.15 | 0.58 | 20.78 | 0.25 | 1.64 | 7.87 | 1912 | 0.14498 |
| GL | 38.47 | 1.19 | 41.73 | 0.65 | 3.25 | 7.79 | 1892 | 0.06000 |
| IB | 5.68 | 0.11 | 5.97 | 0.05 | 0.30 | 4.95 | 1183 | 0.43255 |
| BB | 15.23 | 0.32 | 16.22 | 0.14 | 0.99 | 6.11 | 1470 | 0.19225 |
| LPN | 22.23 | 0.82 | 24.82 | 0.36 | 2.59 | 10.43 | 2567 | 0.12688 |
| LBC | 27.58 | 0.80 | 29.99 | 0.38 | 2.41 | 8.04 | 1953 | 0.09808 |
| LIF | 6.95 | 0.31 | 7.59 | 0.11 | 0.65 | 8.51 | 2074 | 0.32281 |
| AL | 6.44 | 0.11 | 6.64 | 0.04 | 0.20 | 3.06 | 724 | 0.29203 |
| DBC | 13.32 | 0.23 | 13.95 | 0.09 | 0.63 | 4.55 | 1084 | 0.20862 |
| TOT | *NS* | *NS* | *NS* | *NS* | *NS* | *NS* | *NS* | *NS* |
| TAIL | *NS* | *NS* | *NS* | *NS* | *NS* | *NS* | *NS* | *NS* |
| HIND | *NS* | *NS* | *NS* | *NS* | *NS* | *NS* | *NS* | *NS* |
| EAR | *NS* | *NS* | *NS* | *NS* | *NS* | *NS* | *NS* | *NS* |

**Table 3 Linear regressions of the traits of the specimens with the year each specimen was collected.** All 11 traits fit significantly.

|  | ONL | BR | ZB | GL | IB | BB | LPN | LBC | LIF | AL | DBC |
|---|---|---|---|---|---|---|---|---|---|---|---|
| *Multiple R2* | 0.182 | 0.147 | 0.189 | 0.188 | 0.195 | 0.226 | 0.210 | 0.235 | 0.132 | 0.129 | 0.142 |
| *N* | 51 | 56 | 43 | 45 | 58 | 46 | 50 | 45 | 54 | 58 | 42 |
| *F* | 12.136 | 10.505 | 10.787 | 11.179 | 14.836 | 14.118 | 14.059 | 14.531 | 9.030 | 9.445 | 7.796 |
| *P* | 0.001 | 0.002 | 0.002 | 0.002 | <0.0005 | 0.001 | <0.0005 | <0.0005 | 0.004 | 0.003 | 0.008 |
| Constant | −70.193 | −22.431 | −50.736 | −130.835 | −10.938 | −30.063 | −94.951 | −88.491 | −24.662 | −5.719 | −10.143 |

possible explanations for temporal variation of phenotypic characters are: (1) nongenetic, environmental effects (plasticity); (2) gene flow from morphologically different source populations; (3) stochastic evolutionary change through genetic drift; or (4) response to natural selection.

We feel that non-genetic environmental factors such as nutrition or maternal effects are an unlikely explanation for all of the morphological changes observed. First, changes appear so great that some genetic component must be involved. Second, all of the rat's cranial measures increased in size, while at largely the same time most of the mice's traits decreased in size. If environmental plasticity were the cause of the changes we would expect both species' traits to change in the same direction.

Several factors argue against gene flow in the case presented here. First, the rats inhabit oceanic islands separated by at least several kilometers from other islands or from the nearest mainland point. Second, there is no record of rats on nearby islands.

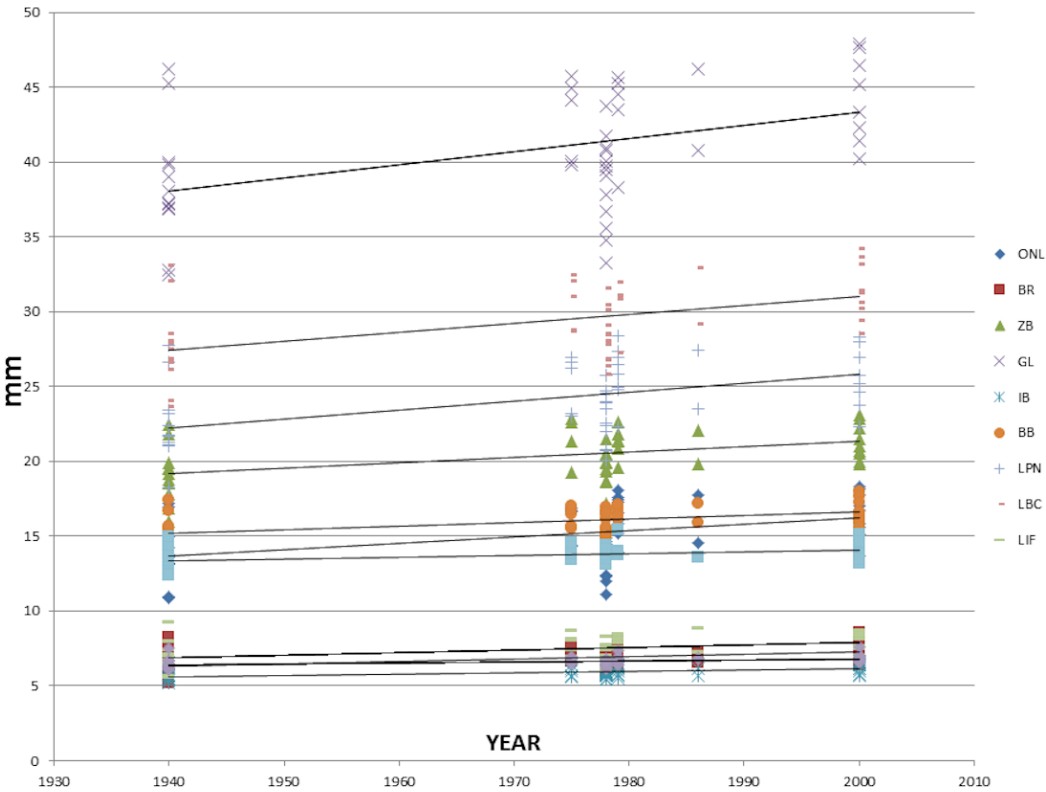

**Figure 1** Linear regressions comparing each measurement of each trait with the year in which the specimen was collected.

A third causative factor that merits consideration is genetic drift. Anacapa is quite small (2.9 sq km, all Anacapa islets together), but mice are often extremely abundant. A three to four year population size cycle is thought to exist on Anacapa, but the minimum size is still probably in the thousands (*Pergams, Lacy & Ashley, 2000*). A final argument against genetic drift is the observation that in both rats and mice many morphological characters are changing, and in most cases changes are in the same direction. Even if certain characters exhibited higher levels of evolutionary plasticity, genetic drift would not be expected to direct changes in many such characters in a uniform direction.

Over approximately the same time frame that some rat traits were getting bigger, endemic deer mice were mostly getting smaller, except that deer mouse ear length and nostral width got bigger (*Pergams & Ashley, 1999*; *Pergams & Ashley, 2000*; *Pergams & Ashley, 2001*). The changes in these mice do not seem to be due to climate change or changes in human population density (*Pergams & Lawler, 2009*), factors which were evaluated in that paper. The changes toward smaller size in the mice are also not consistent with release of selective pressures upon introduction.

We speculate that changes towards mostly smaller size in mice may have been driven by competition with and predation by black rats. *Collins, Storrer & Rindlaub (1979)* study stomach content of both species, and find that both eat at least seven of the same plants: sand lettuce (*Dudleya caespitosa*), wild cucumber (*Echinocystis lobate*), coastal prickly

pear (*Opuntia littoralis*), sea fig (*Carpobrotus chilensis*), iceplant (*Carpobrotus edulis*), slender-leaved iceplant (*Mesembryanthemum nodiflorum*), and holly-leafed cherry (*Prunus ilicifolia*). Although rats do not eat giant coreopsis (*Coreopsis gigantean*) as mice do, rats and mice both employ this large-leaved plant as shelter and cover and so may compete for it. Direct predation of rats upon mice is also a consideration: rats on Anacapa prey on mouse pups, as well as on mice caught in traps (*Collins, Storrer & Rindlaub, 1979*).

If rats were a factor in mouse evolution on Anacapa, it would be interesting to evaluate mouse morphology after the rat eradication in 2001–2002. Such future research might sample Anacapa mice periodically (perhaps every 5–15 years) and begin ongoing analysis after perhaps 20–30 years. If mice were generally getting bigger again, it would further support the hypothesis of competition with and/or predation by rats as a proximate cause of mouse evolution on Anacapa Island.

## ACKNOWLEDGEMENTS

We gratefully thank the museums that provided us access to collections and related support, especially the Field Museum, the Santa Barbara Museum of Natural History, and the Natural History Museum of Los Angeles County. We thank Island Conservation for specimens. We thank Larry Heaney, Bill Stanley, Bruce Patterson, John Phelps, Paul Collins, Kate Faulkner, Cathy Schwemm, Dirk Rodriguez, and Bernie Tershy for advice. DB thanks Melissa Trombley-Byrn and ORWP thanks Valerie Morrow for their personal support.

### Funding

This work was partially supported by National Science Foundation Grant CHE 0629174. The funders had no role in study design, data collection and analysis, decision to publish, or preparation of the manuscript.

### Grant Disclosures

The following grant information was disclosed by the authors:
National Science Foundation Grant: CHE 0629174.

### Competing Interests

The authors declare there are no competing interests.

### Author Contributions

- Oliver R.W. Pergams conceived and designed the experiments, performed the experiments, analyzed the data, contributed reagents/materials/analysis tools, wrote the paper, prepared figures and/or tables, reviewed drafts of the paper.
- David Byrn, Kashawneda L.Y. Lee and Racheal Jackson performed the experiments, wrote the paper, reviewed drafts of the paper.

## Supplemental Information

Supplemental information for this article can be found online at http://dx.doi.org/10.7717/peerj.812#supplemental-information.

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
