# Peer review of "Rapid morphological change in black rats (Rattus rattus) after an island introduction"

_PeerJ, doi:10.7717/peerj.812_

## Round 0.1 · original submission · Major Revisions

· Academic Editor

Major Revisions

The reviewers agree that is an interesting data set and analyses.

Nevertheless the three reviewers had different questions that need to be addressed. In particular they asked for more information and analysis for the two sexes and some other important comments on the statistical analyses.

Please, pay special attention to the comments of Reviewer 3 in relation to the sizes of rats in mainland. How are the sizes in the coast of California? The rats also changed of sizes in the continental populations?

Reviewer 2 asked for a graphical presentation of the data, and I agree this would make the paper more interesting and attractive.

I would also like a better presentation in Methods in relation to the interpretation of the units of measure of change (Darwins and Haldanes), an explicit description of them in Results, and better and clear comparisons of both indices in the Discussion, in particular Haldane’s, that are not mentioned in the Discussion.

Please, attend all the comments of the reviewers in the next version of the paper.

·

Basic reporting

The manuscript is well written and include sufficient introduction and background to demonstrate how the work fits into the broader field of knowledge.

The structure of the submitted manuscript conforme to author guidelines. Nevertheless, there are some details that should be addressed:

Author should use sequential line numbers as stated in the guidelines for authors.

Check for correct in-text citation format: Throughout the text quotes are formatted as Mayr 1963 and should appear as Mayr, 1963. Standardize citation format, and format according to author guidelines.
Check for correct format for the reference section. Journal name should be in italics; book names should be in italics.

Hendry and Kinnison 2003 and Smith et al. 1995 are lacking from the literature cited. Authors should use the symbol "&" insead of "and" in the in-text citations.

Page 4 last paragraph of the introduction: Should “Collins, 1979” be “Collins et al. 1979”? If not, add Collins 1979 to literature cited.

Experimental design

In the Introduction, the authors state the prediction that if the rats from Anacapa have undergone rapid evolutionary change, this could shade additional light in the evolution of endemic mice from this island. Authors should explain the assumptions that underlie this prediction. Furthermore, I think authors should redirect their prediction and state clearly whether they expect to find an increase or a decrease in size in the rats from Anacapa Island. Since the main objective of this study is to determine if rats from Anacapa Island have undergone rapid evolutionary change since their introduction to the island.

In the methods section, authors mention they used measures from 22 males, 16 females and 21 individuals of unknown sex. Nevertheless, the authors don’t explain whether there is sexual dimorphism in this species and if merging data from all individuals in their analyses, regardless of their sex, will introduce bias.

In addition, authors divided their collected specimens in 2 or 3 time periods for analysis, but they do not explain the criteria used to determine these categories. Also, is there some bias related to the disparities in sample size for each time period? Why not analyze each year separately?

Authors compared differences between means pre and post 1950. If rats were introduced to the island in the mid 1800s, what criteria were used to define 1950 as a breaking point?

Finally, the authors performed a linear regression analysis with the MEAN for each period of time. Is it statistically sound to perform a linear regression with the mean of the data? Why not use all the data for each year instead of the mean?

Validity of the findings

My main concern regarding the conclusion of rapid evolutionary change in rats of the Anacapa island is that only three of the morphological measures taken by the authors presented significant change throughout the period of time analyzed. This result was based on a linear regression performed with the mean value for each trait for three time periods. By using the mean for each time period, the variance of the data is eliminated from the analysis, therefore introducing an important bias in the analysis. Why not use all the data for each year instead of the mean?

The authors state in the discussion that there is rapid evolutionary change in Rattus ratus from Anacapa Island because of release of mainland selective pressures and/or increase in food resources. Nevertheless, I consider it is important to discuss whether morphological change in this species could relate to causes other than rapid evolutionary change, such as phenotypic plasticity.

Finally, regarding the influence that rats have had in the evolutionary change of endemic mice:
Pergams and Ashley (1999) found a tendency for morphological change (decrease) in endemic mice from three islands from the Channel Islands. They attribute these changes to both environmental and stochastic factors, but they don’t mention the presence of rats in these islands. Morphological change on black rats, on the other hand, was analyzed only in Anacapa Island. Were black rats introduced to the other islands from the Channel Islands? Is it possible that the sole introduction of rats, independently of morphological change in this species, could have promoted evolutionary change in the endemic mice? How can the hypothesis of evolutionary change in mice related to the introduction of rats to the island be proved?

Additional comments

The authors aimed to determine rapid evolutionary change in body size in Rattus ratus, an invasive species from the California Channel Islands. They found an increase in three cranial measures since 1940 to 2000, when black rats were eradicated from the island. They relate this results with previous findings of a decrease in size of Anacapa mice.
The manuscript is fairly well written and very interesting. Nevertheless, I think authors should focus their predictions on the expected morphological change in rats after their introduction to the island, rather than the effect of rat introduction in morphological change in endemic mice. Also, the discussion should focus on wether they found the expected change and if the observed morphological change is indeed a case of rapid evolutionary change or could be explained by other processes.

·

Basic reporting

The study poses and interesting and seldom evaluated consequence of biological invasion in islands. Overall it is well written and the paper tells a story of rodent invasion. It puts forward possible hypothesis as to why changes in morphological features have ocurred. I think the weakest point of the paper is the speculative nature of the discussion. Dont get me wrong, this I think is a means of generating new hypotheses but some are a little far fetched such as the last sentence of the second to last paragraph of the discussion. This can be completely omitted as the external features (ear length) were not even analyzed adequately due to small sample size. I would have liked to see a broader context an linked to other studies not necessarily restricting them to what would happen in Anacapa.

Experimental design

The design is adequate however further explanation I think is necessary in some areas. For example, Table 1 reports the differences between time classes but the methods describes two different time classes one with three time intervals and another with two time intervals. The table does not specify what time interval the authors are referring to. Are they all using the same time intervals? Please expain further. There is also another confounding factor that was not taken into account which is size differences between sexes. Unfortunately the small sample size precludes a better analysis. I would also have preferrred a graph of the significant linear regressions rather than a table.

Validity of the findings

The findings of the paper are both interesting and give insight into the evolution of rats on islands. Even though sample sizes are small (59 total samples) the conclusions and statistical treatment are adequate. The authors corrected as many errors as possible by either retaking measurements, bonferroni corrections for multiple comparisons, and normality tests. Speculation should be kept to a minimum and specially one in my opinion should be eliminated.

Additional comments

First line of introduction. This rapid evolution must also take into account generation time. Please include information in which generation time plays a role in rapid evolution. This point is important because the rates of change that were calculated in the paper include a factor of generation time. It would just make the point of rapid evolution in this species easier to acknowledge.
The first paragraph is "documented" too often please change wording to not sound so repetitive.
When describing the stomach content of plants please specify scientific name as well as author.
Even much is said of island size in the introduction we have no idea how big Anacapa island is and the idea is not retaken in the discussion. As I said before a borader context for the dicussion would I think improve teh manuscript.

Reviewer 3 ·

Basic reporting

In this study the authors aimed to detect rapid morphological change in black rats introduced to Anacapa Island by mid-1800s. They took measurements of 11 cranial and four external characters. They divided the sample size in two (or three) year periods, and show that cranial measurements increased along time. However, external characters do not showed this trend.

Experimental design

The study is based on samples taken during different years since 1940. Measurements of external characters were taken from a smaller sample size. Thus although their statistical analyses seem sound, conclusions are limited for the fragmentary nature of data.

Validity of the findings

As mentioned, the small sample size, the necessary but arbitrary grouping of data, the unbalance between morphological and cranial data of the same specimens impose limitations on interpretation of results. Discussion is highly speculative.

Additional comments

It would be desirable (1) to know the size of rats from the continent along time; if possible, (2) the size of the rats when the introduction occurred; (3) the correspondence of changes in cranial measurements with those in total size; (2) the effect of sexual/age on body size of black rats.

---

## Round 0.2 · accepted · Accept

· Academic Editor

Accept

I think this is an interesting and relevant contribution and that the new version is clearer and more solid, and I am glad that the paper is now accepted.

I deeply appreciate and thank you efforts to answer all the questions, criticism and concerns of the three reviewers.